# Optic Nerve Injury Enhanced Mitochondrial Fission and Increased Mitochondrial Density without Altering the Uniform Mitochondrial Distribution in the Unmyelinated Axons of Retinal Ganglion Cells in a Mouse Model

**DOI:** 10.3390/ijms24054356

**Published:** 2023-02-22

**Authors:** Takahiro Tsuji, Tomoya Murase, Yoshiyuki Konishi, Masaru Inatani

**Affiliations:** 1Department of Ophthalmology, Faculty of Medical Sciences, University of Fukui, Fukui 910-1193, Japan; 2Research Center for Child Mental Development, Kanazawa University, Kanazawa 920-8640, Japan; 3Life Science Innovation Center, University of Fukui, Fukui 910-8507, Japan; 4Department of Applied Chemistry and Biotechnology, Faculty of Engineering, University of Fukui, Fukui 910-8507, Japan

**Keywords:** glaucoma, optic neuropathy, mitochondria, optic nerve crush, confocal scanning ophthalmoscope, distribution, Thy1-mito-YFP mice

## Abstract

Glaucomatous optic neuropathy (GON), a major cause of blindness, is characterized by the loss of retinal ganglion cells (RGCs) and the degeneration of their axons. Mitochondria are deeply involved in maintaining the health of RGCs and their axons. Therefore, lots of attempts have been made to develop diagnostic tools and therapies targeting mitochondria. Recently, we reported that mitochondria are uniformly distributed in the unmyelinated axons of RGCs, possibly owing to the ATP gradient. Thus, using transgenic mice expressing yellow fluorescent protein targeting mitochondria exclusively in RGCs within the retina, we assessed the alteration of mitochondrial distributions induced by optic nerve crush (ONC) via in vitro flat-mount retinal sections and in vivo fundus images captured with a confocal scanning ophthalmoscope. We observed that the mitochondrial distribution in the unmyelinated axons of survived RGCs after ONC remained uniform, although their density increased. Furthermore, via in vitro analysis, we discovered that the mitochondrial size is attenuated following ONC. These results suggest that ONC induces mitochondrial fission without disrupting the uniform mitochondrial distribution, possibly preventing axonal degeneration and apoptosis. The in vivo visualization system of axonal mitochondria in RGCs may be applicable in the detection of the progression of GON in animal studies and potentially in humans.

## 1. Introduction

Mitochondria are intracellular organelles that control cellular metabolism and produce approximately 90% of the cell’s ATP, an essential form of energy to maintain vital functions. They are prevalent in energy-demanding organs, particularly the central nervous system (CNS). Mitochondrial dysfunction is deeply involved in the pathogenesis of neurodegenerative diseases such as Alzheimer’s disease, Parkinson’s disease, Huntington’s disease, and amyotrophic lateral sclerosis. In addition, the retinas are also mitochondria-rich organs as they demand high levels of energy to maintain their neurites and active pulse transmission. Thus, mitochondrial failure causes several types of optic neuropathy, such as Leber’s hereditary optic neuropathy (caused by point mutations in the mitochondrial DNA) and autosomal dominant optic atrophy (ADOA; caused by mutations in the *OPA1* gene, which is involved in mitochondrial fusion) [1].

Glaucomatous optic neuropathy (GON) is a common disease characterized by a loss of retinal ganglion cells (RGCs). The axonal degeneration of RGCs is associated with several factors, such as intraocular pressure (IOP), ocular blood flow, and aging. According to a recent systematic review and meta-analysis, GON was the second most common cause of visual impairment in the world and the most common irreversible cause of blindness in 2020 for people over 50 years of age [2]. Genetic variants in mitochondrial and mitochondria-related genes have been reported in patients with GON [3,4]. OPA1, a mitochondrial fusion-related protein, is responsible for ADOA, and patients with glaucoma reportedly carry variants of this gene [5,6]. OPA1-deficient, aged mice (with heterozygous mutations) also reportedly exhibit defective visual function and axonal degeneration [7,8]. Furthermore, mitochondria accumulate beneath the cribriform plate area of the optic disc in the eyes of several experimental model animals with induced high IOP as well as glaucoma patients [9]. Impaired energy metabolism and elevated ROS levels, caused by mitochondrial dysfunction, lead to the apoptosis of RGCs [10]. The oral administration of vitamin B3, a precursor of nicotinamide adenine dinucleotide (NAD), which can improve mitochondrial function, reportedly prevents the progression of glaucoma in aged mice and improves the function of the inner retinal layer in patients with glaucoma [11,12]. Moreover, the transplantation of fresh mitochondria into the injured optic or sciatic nerve reportedly attenuates axonal degeneration [13,14]. Thus, the status of mitochondria in the damaged nerve might serve as a diagnosis index and a therapeutic target, especially in the optic nerve.

The involvement of mitochondria in glaucoma has been well studied in terms of apoptosis, but not in terms of the maintenance of nerve fibers. Mitochondria accumulate in the growth cones and branches of neurites to provide ATP for their extension [15,16]. In addition, bulb-shaped varicosities exist within the unmyelinated parts of RGC axons in humans and non-human primates. Mitochondria are prevalent in these structures, which may contribute to the transmission of electrical signals in the unmyelinated fibers [17]. We previously demonstrated that mitochondria are uniformly distributed in the unmyelinated parts of RGC axons in transgenic mice with visualized mitochondria. An ATP-dependent mechanism may be responsible for this distribution pattern, as we explored using primary cultured cerebellar granule cells [18]. However, the pathophysiological role of this mitochondrial distribution pattern remains unknown. Thus, in this study, we introduced an optic nerve crush (ONC) model to represent glaucoma or traumatic optic neuropathy. Our aim was to examine how the mitochondrial distribution pattern was altered in the unmyelinated part of the RGC axons by using transgenic mice with yellow fluorescent protein (YFP) expressed exclusively within RGC mitochondria, in vivo and in vitro. We also aimed to determine whether the results were consistent between the in vivo and in vitro examinations. We discovered that the injury did not alter the mitochondrial distribution pattern, although the mitochondria fissioned into smaller ones.

## 2. Results

### 2.1. Intra-Mitochondrial Distribution in a Single Unmyelinated RGC Axon of Thy1-mito-YFP Mice

We generated a transgenic mouse in which mitochondrial localization signals were fused to a YFP under the Thy1 promotor (Thy1-mito-YFP mice), which is specifically expressed in RGCs, as described in detail in Section 4.1. This allowed us to observe the distribution of YFP signals in a single RGC axon in a flat-mounted retinal section (Figure 1a). Images were taken with a confocal microscope to visualize the YFP signals of individual mitochondria in a single RGC axon. Thereafter, we binarized and straightened the signals of a single axon with Fiji ImageJ to measure the position of individual mitochondria (Figure 1b) [19]. We calculated the Iδ index, which indicates a distribution pattern, and plotted its relationship to the compartment width (Figure 1c). We confirmed that the distribution of mitochondria in a single axon of RGCs exhibited a uniform pattern in vivo and in vitro (see control in Figures 3d and 5c), as previously reported [18].

### 2.2. Effects of ONC on Loss of RGCs

We examined whether this pattern could be disrupted in a pathological state. We used the ONC model, simulating traumatic optic neuropathy or a glaucomatous state. We analyzed the distribution pattern, density, and area of mitochondria, as described in Section 4. First, we counted the number of Brn3-positive cells in each flat-mounted retinal section, 1 and 7 days after ONC (Figure 2a,b). While the number of Brn3-positive cells did not differ from the control section 1 day after ONC (to 157.6 ± 11.2/1.9 mm^2^ and 129.7 ± 14.6/1.9 mm^2^, respectively, *n* = 6, *p* = 0.39, Mann–Whitney U test), it had decreased more 7 days after ONC than that in the control section (to 29.8 ± 6.0/1.9 mm^2^ and 155.9 ± 7.3/1.9 mm^2^, respectively, *n* = 5, *p* = 0.0079, Mann–Whitney U test) (Figure 2c). Thus, we successfully established the ONC model to induce RGC apoptosis.

### 2.3. Effects of ONC on the Mitochondrial Distribution, Density, and Area in a Single RGC Axon

We analyzed the mitochondrial distribution of a single axon, 1 and 7 days after ONC (Figure 3a–c). We calculated the Iδ index and plotted that against the compartment width. The graph formed a sigmoid curve approaching 1 for control segments and those 1 and 7 days after ONC, indicating a uniform distribution pattern (Figure 3d). These results revealed that ONC did not affect mitochondrial distribution.

Subsequently, we measured the density and area of mitochondria in a single axon. The density of mitochondria 1 day after ONC was similar to that in the control axon (1.26 ± 0.08/μm (*n* = 13) and 1.16 ± 0.04/μm (*n* = 14), respectively, *p* = 0.69, Mann–Whitney U test), while that 7 days after ONC was higher than that in the control axon (1.50 ± 0.05/μm (*n* = 25) and 1.24 ± 0.04/μm (*n* = 29), respectively, *p* = 0.0003, Mann–Whitney U test) (Figure 3e). We also measured the area of mitochondria 1 and 7 days after ONC. The area of mitochondria 1 day after ONC was not different from that in the control axon (4.36 ± 0.19 μm^2^ (*n* = 263) and 4.29 ± 0.22 μm^2^ (*n* = 244), respectively, *p* = 0.39, Mann–Whitney U test), whereas that 7 days after ONC was smaller than that in the control axon (2.13 ± 0.07 μm^2^ (*n* = 570) and 3.46 ± 0.14 μm^2^ (*n* = 528), respectively, *p* < 0.0001, Mann–Whitney U test) (Figure 3f). We generated a frequency histogram of the area of the mitochondria, revealing some difference in the graph between the control axon and 1 day after ONC (kurtosis: 4.5 and 0.7, skewness: 1.7 and 0.9, control and 1 day, respectively) but, particularly noteworthy, higher kurtosis in the graph 7 days after ONC than that of the control axon (kurtosis: 6.0 and 23.5, skewness: 2.1 and 3.3, control and 7 days, respectively). Thus, ONC decreased the number of larger mitochondria and increased the number of smaller mitochondria.

### 2.4. In Vivo Imaging of YFP Signal Located in Mitochondria within RGCs

The YFP signal can be captured by excitation with light at a wavelength of 490 nm via a commercially available confocal scanning laser ophthalmoscope (cSLO; F10, Nidek, Okazaki, Japan). We generated a transgenic mouse that expresses YFP exclusively in the mitochondria of Thy1-positive cells (Thy1-mito-YFP mice). Thy1 is a cell-surface glycoprotein and, in the adult retina, is mainly expressed in RGCs. We performed fundus imaging of the transgenic mice in vivo to capture mitochondria in RGCs, especially within a single, unmyelinated axon. We detected fluorescent signals and measured the total fluorescence intensity in the fundus, possibly reflecting the total volume of mitochondria in RGCs (Figure 4a,b). Moreover, we detected the fluorescence signal in individual images along a single axon of RGCs (Figure 5a,b). Thus, we could analyze the total expression of mitochondria in RGCs and the density and location of each mitochondrion within a single axon. However, the limitations of this method are the inability to measure the area of individual mitochondria and to track them along the same region of the axon over several days. The reason for the former is that the signals are subject to halation, prohibiting the binarization of the images. The reason for the latter is that the area of sufficient focus is limited, and the process of finding the same area is too time-intensive. As time progresses after mice are anesthetized, their lenses become cloudy. Therefore, all imaging should be completed within approximately 10 min.

### 2.5. Validation of Change in the Intensity of YFP in the Fundus of Thy1-mito-YFP Mice after ONC

We took images of the fundus in five directions (frontal, superior, inferior, auricular, and nasal) with cSLO 0, 1, 3, and 7 days after ONC. Thereafter, we measured the fluorescence intensity of the same areas in the four quadrants (top, bottom, left, and right), approximately two optic disc diameters from the optic disc head (Figure 4a,b). The average intensity of these four areas was calculated, as was the intensity ratio, by dividing the intensity at each time point by the basal intensity before ONC (at 0 days). Repeated-measures one-way analysis of variance (RM one-way ANOVA) indicated a difference in the intensity ratio among days (F (1.849, 14.79) = 19.79, *p* < 0.0001), and post hoc analysis via Tukey’s multiple comparisons test revealed differences at 3 (*p* = 0.0127) and 7 days (*p* < 0.0001) compared to 0 days but not at 1 day after ONC (*p* = 0.5087) (Figure 4c). Moreover, we also discovered significant differences between 1 and 7 days (*p* = 0.0059) and 3 and 7 days (*p* = 0.0351) (Figure 4c). On the other hand, we did not observe any effect of sham surgery in the fellow eye on the fluorescence intensity (F (2.454, 9.817) = 1.453, *p* = 0.2840, compared among days, RM one-way ANOVA) (Figure 4c). These data suggest that the fluorescence intensity decreased as days passed, consistent with the results of flat-mounted retinal sections.

### 2.6. Effects of ONC on Mitochondrial Distribution and Density in a Single RGC Axon

As it was not feasible to focus on the exact same location in each axon on subsequent days, we randomly focused on as many axons as possible in each quadrant and determined the location of the signal within the single axons (Figure 5a). In further experiments, we performed ONC on the right eye and sham-operated the left eye as a control (sham operation). We measured the location within a single axon, 1 and 7 days after ONC. The distribution of mitochondria was determined with the Iδ index, as described later in Section 4.5. As reported in Section 2.1 and Section 2.3 for flat-mounted retinal sections, the graph of the Iδ index of the control axons against the compartment width in the in vivo fundus imaging with the cSLO also revealed a uniform pattern (Figure 5c). One and seven days after ONC, the graphs of the Iδ index against the compartment width similarly displayed little difference compared to the control axons (Figure 5c). Thereafter, we measured the density of mitochondria in a single axon. One day after ONC, the density was unchanged compared to the control axon, but seven days after ONC, the mitochondrial density was higher than that in the control axon (Figure 5d). Thus, we revealed that the mitochondrial distribution was not disturbed by ONC and that the mitochondrial density increased with ONC, as observed in the in vitro experiment with flat-mounted retinal sections.

### 2.7. Comparison of Mitochondrial Images in Axons between In Vivo and In Vitro Imaging

To verify that the mitochondrial distribution and density did not differ between the in vivo and in vitro imaging methods, we compared the distribution of mitochondria within a single axon of an RGC observed in in vitro flat-mounted retinal sections to that within the same axon observed with in vivo cSLO fundus imaging. After in vivo cSLO fundus imaging, the direction of the eye was marked and flat-mounting was performed, as described in Section 4. The site of a single axon of interest captured with in vivo imaging was identified by using its position relative to the optic disc and the blood vessels (Figure 6). The identified axon was observed with a confocal microscope in the flat-mounted retinal section. Both images were processed to analyze the mitochondrial distribution with Fiji ImageJ. We marked each mitochondrion in the linearized single axon in both images and revealed that they were similarly distributed (Figure 6). These results demonstrated the reliability of the analysis of mitochondrial distribution via in vivo cSLO fundus imaging.

## 3. Discussion

In this study, we demonstrated the way in which the mitochondrial distribution was altered within a single RGC axon in Thy1-mito-YFP mice, both in vitro and in vivo. We previously reported that the distribution of mitochondria in the axons of both RGCs and primary cerebellar granule cells displayed a uniform pattern [18]. To reveal the pathophysiological role of these distributions, we introduced an ONC model to simulate a traumatic optic neuropathy or a type of glaucoma. One and seven days after ONC, the uniform mitochondrial distribution was not disrupted, and seven days after ONC, the density of mitochondria was increased while their size became smaller in the in vitro flat-mounted retinal sections. Furthermore, we developed an in vivo imaging system, using a cSLO, which allowed us to monitor mitochondria in the fundus in vivo over several days. Thereby, we confirmed the results of the in vitro study.

### 3.1. Pathophysiological Significance of Mitochondrial Distribution in an ONC Mouse Model

Mitochondria accumulate in growth cones and at branches of nerve fibers during development, as well as at the distal tips of damaged axons [15,16]. On the other hand, axonal mitochondria are distributed at regular intervals in dorsal root ganglions (DRGs), cerebellar granule cells, and RGCs [18,20]. Considering that mitochondria can act to buffer ATP and calcium, this regular distribution of axonal mitochondria may be involved in axonal maintenance. In this study, we introduced an ONC mouse model and examined the distribution and size of mitochondria in the unmyelinated parts of the axons, both in vivo and in vitro. As the axonal mitochondria decreased in size, their distribution remained uniform, probably contributing to the prevention of axonal degeneration.

Axotomy to primary DRGs from mice causes damage to the axon cell membrane, the degeneration of the cytoskeleton, and the depolarization of mitochondria [21]. Mitochondrial dysfunction causes a decrease in local ATP production, the opening of the mitochondrial permeability transition pore, and the release of cytochrome c into the cytosol, resulting in axonal degeneration. On the other hand, an increase in CCAAT/enhancer-binding homologous proteins, a marker of endoplasmic reticulum stress, occurs in the cytosol of RGCs by their axonal injury, resulting in their apoptosis [22,23]. We discovered an approximately 80% reduction in the number of RGCs 7 days after ONC with survived RGC unrecovered fission of axonal mitochondria. On the contrary, a peripheral spinal cord injury reportedly induces the fission of axonal mitochondria in the first week after the injury, but their size recovers to the basal level in the second week [24]. This difference in mitochondrial changes may resemble the different regenerative capacities between the CNS and the peripheral nervous system (PNS), thus resisting their apoptosis in the PNS. In fact, the suppression of Drp1, a protein involved in mitochondrial fission, or the overexpression of OPA1, a protein involved in mitochondrial fusion, protects RGCs and prevents their axonal degeneration in a mouse glaucoma model [25,26]. Thus, much research may be necessary concerning the distinct capacity of mitochondrial length and area to recover after nerve injury in the PNS and CNS.

Although RGC axonal mitochondria were smaller after ONC, their distribution remained uniform. However, we could not determine whether mitochondria fissioned and remained at a site or fissioned and dispersed before settling into a uniform distribution. Mitochondria in both axons and dendrites reportedly migrate to a lesser degree than cultured mammalian neurons that are matured in vitro [27,28]. Small mitochondria are reportedly more motile in primary embryonic chick DRGs than large mitochondria [20]. Considering these results, fission may be a necessary process for the redistribution of mitochondria to sites with high energy demands, e.g., the distal tips of injured axons as well as the maintenance of long axons.

### 3.2. Evaluation of Optic Nerve Disease via the Intra-Axonal Mitochondrial Imaging of RGCs

As mitochondrial dysfunction is closely related to the onset and extension of GON, its clinical application has been examined. The intensity of autofluorescence, representing oxidative stress in mitochondria, is reportedly higher in patients with open-angle glaucoma and ocular hypertension than that in controls [29,30,31]. Using albino mice expressing cyan fluorescent protein specifically in the mitochondria of RGCs, Takihara et al. developed a visualizing system to analyze mitochondrial dynamics in RGC axons through the sclera [32]. They reported that mitochondrial transport in both directions was attenuated 3 days after inducing elevated IOP, which may be an index of glaucoma progression. One of the limitations of this study was that the area of observation was small in the peripheral areas of the retina and that repeated experiments to observe the same areas of focus over time were not feasible. Mitochondria fissioned in the axons beneath optic nerve papilla in DBA2/J mice and high-IOP model mice, as well as in the axons of cultured RGCs with elevated hydrostatic pressure [33,34]. Thus, reduced mitochondrial motility and an enhanced level of fissioning may indicate the degree of RGC damage and reflect the severity of optic neuropathy. In the present study, we observed mitochondrial fissioning in RGC axons after ONC, along with an enhanced density of mitochondria in vitro in flat-mounted sections. We could also analyze the mitochondrial density in the axons after ONC via in vivo imaging with cSLO, but not mitochondrial motility and size because of the low resolution of fundus images. This in vivo imaging system may be clinically useful in cases of optic neuropathy, including glaucoma. However, the positive relationship between mitochondrial fission and density should be verified, and higher-resolution fundus images need to be developed to enable the analysis of mitochondrial size and, ultimately, motility.

### 3.3. Limitations of the Present Method

In the current study, we examined whether the distribution pattern and density of mitochondria is an indicator of optic neuropathy. Although we assumed that all observed mitochondria were immobile, we could not distinguish between mobile and immobile mitochondria with our present methods. In in vivo research, more than 90% of axonal mitochondria in the cortical pyramidal layer are reportedly immobile, as are more than 99% of those in V1 of the visual cortex [27,35]. Similarly, only 7.2% of mitochondria within the axons of optic nerve explants of mice under 6 months old are mobile [36]. On the other hand, 72.4% of axonal mitochondria in primary RGCs are reportedly mobile during the early days in vitro (DIV), much higher than that of other primary neurons [37]. Moreover, 70% to 90% of axonal mitochondria in primary cortical neurons were immobile at 7 DIV and 98% at 28 DIV [27]. Considering these results and the higher energy demand of the retina, we cannot exclude the possibility of higher mitochondrial motility in the unmyelinated parts of RGCs.

## 4. Materials and Methods

### 4.1. Animals

The transgenic mice in which YFP with human cytochrome c oxidase subunit VIII were expressed under the regulatory element of the mouse Thy1 gene (Thy1-mito-YFP mice) were generated as previously described [18]. In total, we used 36 of these male mice at 12 to 16 weeks old. In the retina, these mice expressed the YFP exclusively in the mitochondria of RGCs. Animals were kept at a temperature of 20–26 °C, a humidity of 40–60%, and a 12 h light and 12 h dark cycle and ad libitum feeding. The experiment was carried out in accordance with the Association for Research in Vision and Ophthalmology Statement for the Use of Animals in Ophthalmic and Vision Research and approved by the Committee on Animal Experimentation of Fukui University (approval number: R3069).

### 4.2. ONC

Animals were anesthetized with an intraperitoneal injection of a 5 mL/kg mixture of medetomidine (0.3 mg/kg; catalog no.: 987685986, Nippon Zenyaku Kogyo, Koriyama, Japan), midazolam (4 mg/kg; catalog no.: 211762100, Astellas Pharma, Tokyo, Japan), and butorphanol (5 mg/kg; catalog no.: 222866132, Meiji Seika Pharma, Tokyo, Japan) in saline. Immediately after losing their righting reflex, animals were topically administered a 0.4% oxybuprocaine hydrochloride ophthalmic solution (Santen Pharmaceutical Co., Ltd., Osaka, Japan) for local anesthesia. An incision of the conjunctiva was made at the upper temporal limbus, and connective tissue and rectus muscles were gently removed to access the posterior globe and the optic nerve without damage to the vortex vein. The optic nerve was pinched approximately posterior to the globe with curved forceps for 10 s. After the addition of a small amount of ofloxacin ophthalmic ointment (Santen Pharmaceutical Co., Ltd., Osaka, Japan), atipamezole, an antagonist of medetomidine, (3 mg/kg, Nippon Zenyaku Kogyo Co., Ltd., Fukushima, Japan) was intraperitoneally injected to confirm the recovery of the righting reflex.

### 4.3. In Vivo CSLO Imaging and Data Analysis

A commercially available cSLO with a wide-viewing lens (cSLO; F10, NIDEK CO., LTD, Nagoya, Japan) was applied to the ocular surface after cutting the eyelashes, and after treating the ocular surface with a hydroxyethyl cellulose eye solution (Senju Pharmaceutical Co., Ltd., Osaka, Japan) to keep moisture, a contact lens (0 diopters, 1.70 mm in radius, 3.20 mm in diameter, and polymethyl methacrylate in material, Micro-M, Cantor and Nissel Ltd., Brackley, UK) was mounted on the cornea following the removal of the excess liquid. The animals were positioned on a hand-made platform next to the camera lens. Twenty images of 600 × 800 pixels in size were automatically captured in succession per 20 Hz frame rate for every shoot. Immediately after the experiment, the animals were recovered, as described above. The instruments showed the pixel size, not the actual length.

The arrangement of all of the photos was processed via the stack-reg plugin of the Fiji ImageJ version 1.54b software (National Institute of Health, Bethesda, MD, USA). The arranged photos were stacked to make one image. The photos were taken from five different directions (center, upper, temporal, nasal, and lower) with the same light intensity and aperture. Figure 1 is an automatically compiled panorama image made of four or five photos, made in Adobe Photoshop Elements 2022 version 20.0 (Adobe Inc., San Jose, CA, USA). All photos were taken from 2 to 5 o’clock in the afternoon. The fluorescence intensity of the 200 × 200-pixel areas in each of the four quadrants, arranged approximately two optic disc diameters from the optic disc head, was measured in the stacked panorama image with ImageJ. The same area was measured on each experimental day for comparative purposes.

To analyze the mitochondrial distribution, a single axon that did not overlap with other axons was randomly selected from within the fundus picture. Comparisons between groups in the in vivo imaging were not conducted in a blinded manner; rather, all axons in which mitochondrial dots could be clearly drawn via particle analysis in ImageJ were mechanically analyzed in each group. We could measure the location but not the size of the mitochondria, because each dot was too blurred.

### 4.4. Flat-Mounted Retinal Section and Immunohistochemistry

Mice were overdose anesthetized and fixed with transcardial perfusion with 4% paraformaldehyde in phosphate-buffered saline (PBS) following a flush of PBS. The eyes were post-fixed for another hour and the retinas were dissected. They were rinsed with PBS and flattened by making four to five radial cuts. Immunohistochemistry for Brn3 proceeded with a method modified from that previously described [32]. Briefly, the retinas were permeabilized with 0.5% triton X-100 in PBS for 1 h, blocked with blocking solution (10% normal goat serum (catalog no.: 005-000-121, Jackson ImmunoResearch Laboratories, West Grove, PA, USA)), and incubated with anti-Brn3 antibody (1:50, catalog no.: sc-6026, Santa Cruz Biotechnology Inc, Dallas, TX, USA) diluted in the blocking solution. After washing the retinas with PBS, they were incubated with goat anti-mouse IgG H&L (Alexa Fluor 594) preadsorbed (catalog no.: ab150120, Abcam PLC, Cambridge, UK) and diluted in 0.5% triton X-100 in PBS. They were washed with PBS, mounted on glass slides, and cover-slipped (catalog no.: TA-006-FM, Thermo Fisher Scientific, Waltham, MA, USA).

### 4.5. Imaging of Whole Retinal Mounts and Data Analysis

To count Brn3-positive cells, the images were taken with an upright microscope (BX51, Olympus, Tokyo, Japan) equipped with a charged-coupled device camera using a 20× objective lens at a resolution of 1024 × 1024 pixels. The images were taken in three different areas (proximal, middle, and peripheral) in each of the four quadrants. The number of Brn3-positive cells was counted manually with ImageJ in a manner blind to the examiner.

YFP signals on the axons were observed with a confocal microscope (FV1200, Olympus, Tokyo, Japan), and images were acquired at a resolution of 1024 × 1024 pixels by using 10×, 20×, and 60× objective lenses at a site close to the ora serrata, because the axons were sparse enough to allow the identification of a single axon and the analysis of the mitochondrial location and size along the axon. The location and size of each mitochondrial dot obtained with the 20× objective lens were analyzed via particle analysis with ImageJ software, as described above. These images were clearer than those taken with the F10 cSLO, and they enabled the measurement of mitochondrial size. The distribution of mitochondria was determined with the *Iδ* index, which is used to measure the dispersion of each individual in a group [38]. The formula for the *Iδ* index was derived from an equation that involves the number of compartments (*q*) and the mitochondrial count in each compartment (*x_i_*), as previously described [18].
$$I\delta =q\overset{q}{\underset{i=1}{\sum }}x_{i}\left(x_{i}-1\right)/\overset{q}{\underset{i=1}{\sum }}x_{i}(\overset{q}{\underset{i=1}{\sum }}x_{i}-1)$$

### 4.6. Statistical Analysis

Statistical analysis was performed using Prism 8 software (GraphPad Software Inc., San Diego, CA, USA). The data are presented as means ± standard errors of the mean. RM one-way ANOVA was used to assess the effect of ONC on YFP signals in the mitochondria, followed by post hoc analysis with Tukey’s multiple comparisons test. In the rest of the experiments, we used the left eye as the control (sham-operated) and the right eye as the ONC model. Mann–Whitney U tests were used to assess the effects of ONC on the counts of Brn3-positive cells and the density and dot size of mitochondria. 

## 5. Conclusions

In conclusion, optic nerve injury induced the fission of mitochondria without altering their distribution within the globe part of the RGC axon. The in vivo imaging of RGC axonal mitochondria with a cSLO may contribute to the assessment of patients with optic neuropathy and reveal the role of mitochondrial dynamics on the apoptosis of RGCs and their axonal degeneration in patients with optic neuropathy, including glaucoma.

## Figures and Tables

**Figure 1 ijms-24-04356-f001:**
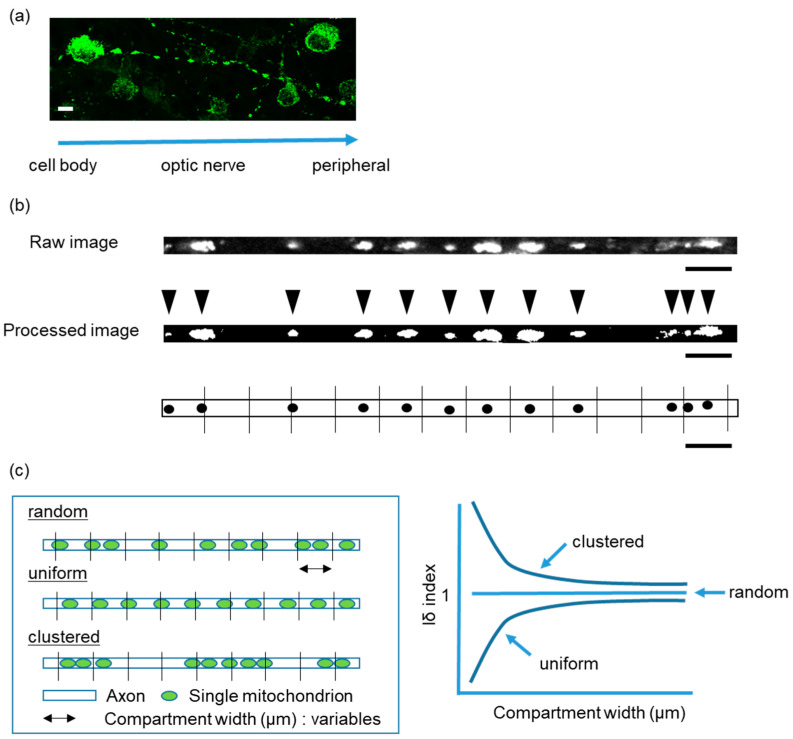
(**a**) Example of visualized mitochondria in a single axon of a retinal ganglion cell. Mitochondria (YFP) shown in green. Bar, 10 μm. (**b**) The original image (top) and the binarized image (middle) of mitochondria processed in a linearized axon. Arrowheads indicate a single mitochondrion each. The bottom panel is a schema of the mitochondrial distribution in each 10 μm compartment. Black dots shown as the position of each mitochondrion. Bar, 10 μm. (**c**) Pattern of mitochondrial distribution in a single axon (left), and the relationship of the Iδ index to the compartment width (right). Green dots indicated as single mitochondrion. The Iδ index against each compartment width is calculated with the formula (see Section 4.5). The clustered, random, and uniform patterns are indicated in both panels (modified Figure 2 in [18]).

**Figure 2 ijms-24-04356-f002:**
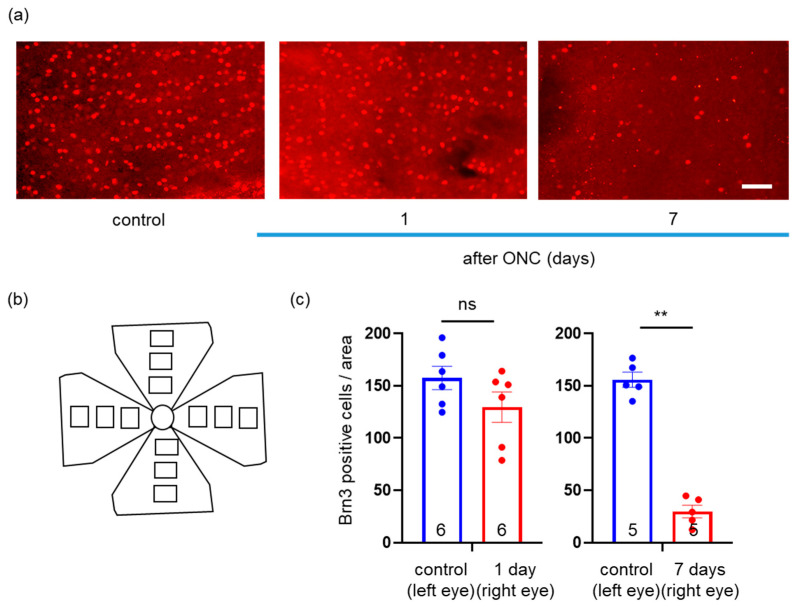
(**a**) Immunohistochemistry of Brn3 in the flat-mounted retinal section. Left panel: control; middle panel: 1 day after ONC; right panel: 7 days after ONC. Brn3 protein shown in red circluar staing in nucleus. Bar, 100 μm. ONC: optic nerve crush. (**b**) Schema of the areas (1593.75 μm × 1200 μm) in which we counted Brn3-positive cells. The areas (blocks) were the proximal (250 μm from the optic nerve head), middle (800 μm from the optic nerve head), and peripheral (100 μm from the ora serrata) areas of each quadrant. (**c**) The average counts in the 12 areas per retina. The number of retinas used in each group for quantification is indicated in the graphs. Data are presented as the mean ± SEM. ** *p* < 0.01, ns = not significant.

**Figure 3 ijms-24-04356-f003:**
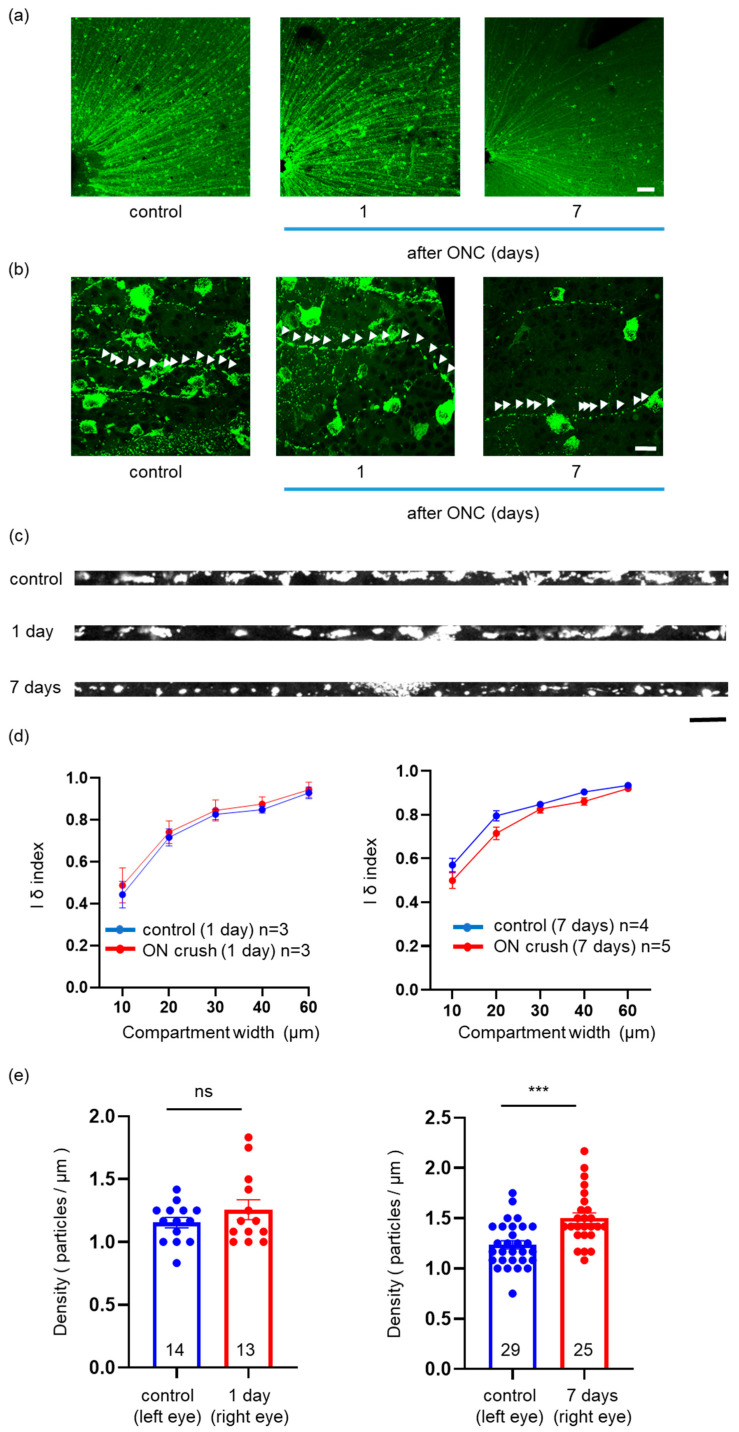
Visualization of YFP signals in the flat-mounted retinal section and mitochondrial distribution. (**a**) Image at low magnification. Mitochondria (YFP) shown in green. Bar, 200 μm. Left panel: control; middle panel: 1 day; and right panel: 7 days after ONC. (**b**) Image at high magnification. Mitochondria (YFP) shown in green. Bar, 20 μm. Each arrowhead indicates a single mitochondrion within a single axon. Left panel: control; middle panel: 1 day; and right panel: 7 days after ONC. (**c**) Mitochondria visualized within a linearized axon. Mitochondria (YFP) shown in white. Bar, 10 μm. Upper panel: control; middle panel: 1 day; and lower panel: 7 days after ONC. YFP: yellow fluorescent protein; ONC: optic nerve crush. (**d**) Plotting of the Iδ index against compartment width. All the graphs indicate that the pattern of mitochondrial distribution was uniform. The number of retinas used for quantification in each group is indicated in the graphs. Data are presented as the mean ± SEM. (**e**) The average density of mitochondria in the axons. The number of axons used for quantification in each group is indicated in the graphs. Data are presented as the mean ± SEM. *** *p* < 0.001, ns = not significant. (**f**) The average area of mitochondria in the axons. The number of mitochondria used for quantification in each group is indicated in the graphs. Data are presented as the mean ± SEM. **** *p* < 0.0001, ns = not significant. (**g**) Frequency distribution of the average area of the mitochondria in the axons.

**Figure 4 ijms-24-04356-f004:**
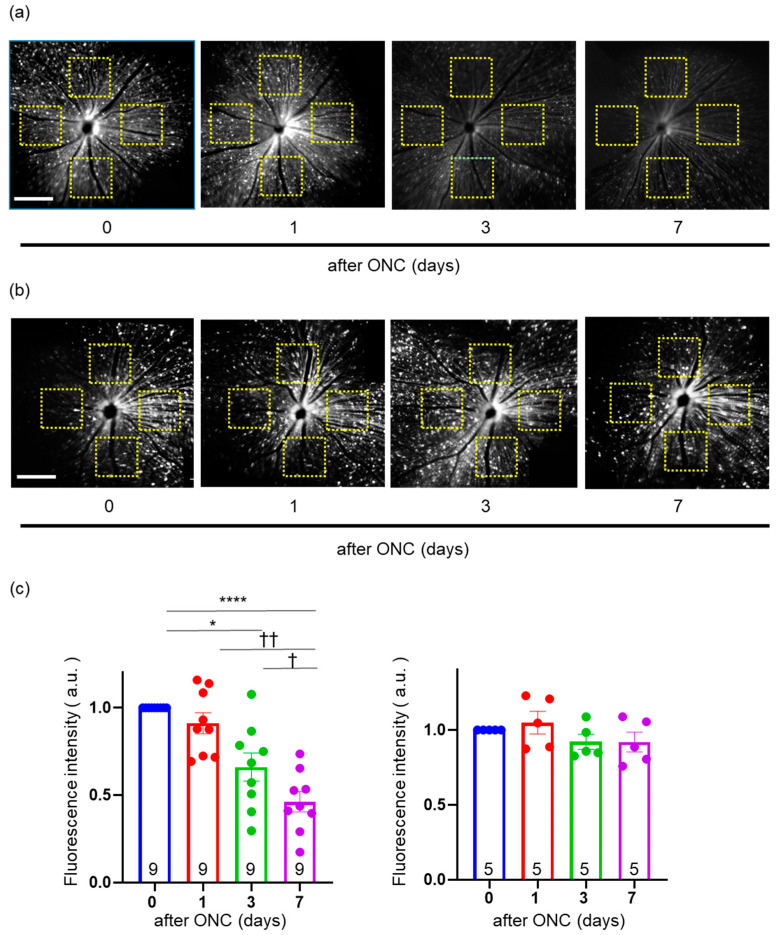
(**a**) Fundus image of the ONC-operated eyes observed with a cSLO. The intensity was measured in the yellow squares (200 × 200 pixels) in four quadrants, each two diameters of the optic disc apart from the optic disc head. Mitochondria (YFP) shown in white. Bar, 200 pixels. (**b**) Fundus image of the sham-operated eyes (control eyes) observed with cSLO. cSLO: confocal scanning laser ophthalmoscope; ONC: optic nerve crush. (**c**) Ratio of the average fluorescence intensity to the basal intensity (at 0 days). The number of retinas used for quantification in each group is indicated in the graphs. au, arbitrary unit. Left panel: ONC-operated group; right panel: sham-operated (control) group. Data are presented as the mean ± SEM. * *p* < 0.05, **** *p* < 0.0001, † *p* < 0.05, †† *p* < 0.01.

**Figure 5 ijms-24-04356-f005:**
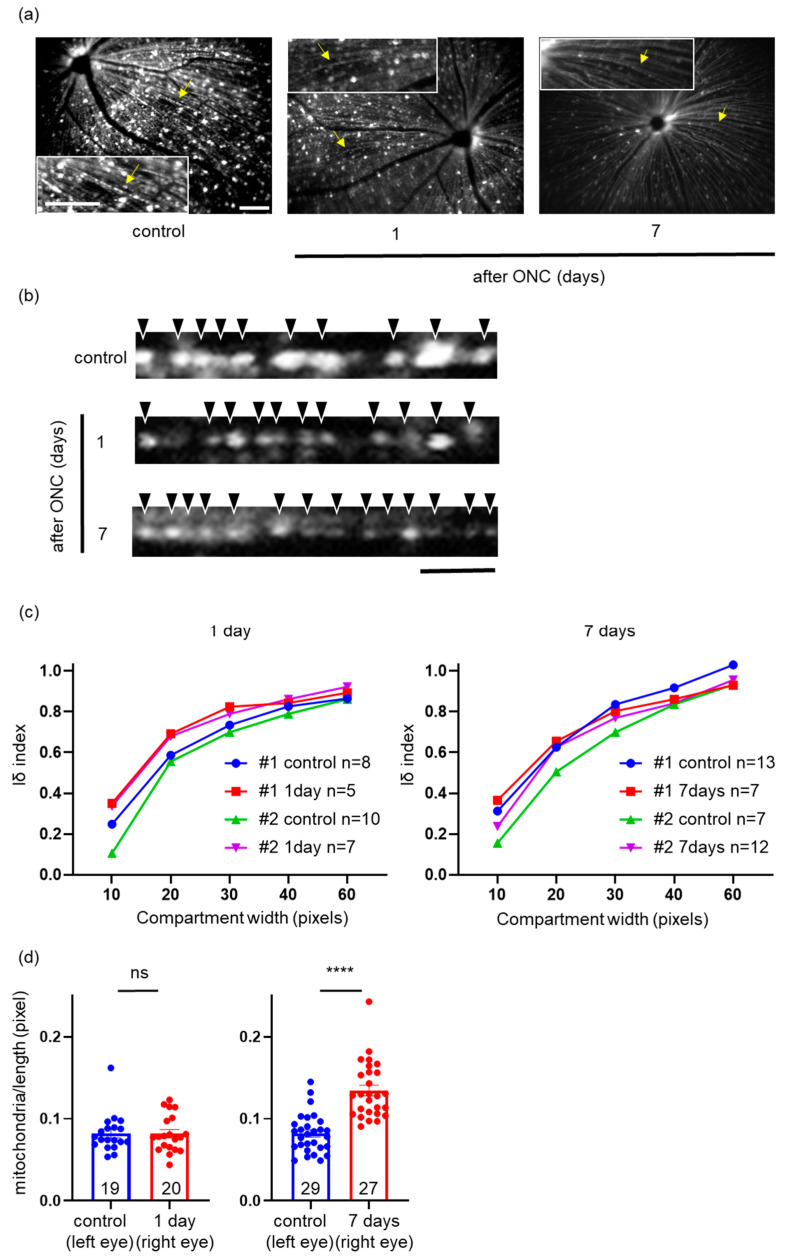
In vivo cSLO imaging of mitochondria within a single axon and mitochondrial distribution. (**a**) The original fundus images of yellow fluorescent protein. Enlarged images are provided in the insets. Yellow arrows indicate the analyzed axons in (**b**). Mitochondria (YFP) shown in white. Bars, 100 pixels. (**b**) Mitochondria visualized within a linearized axon captured with a cSLO. Black arrowheads indicate individual mitochondria. Mitochondria (YFP) shown in white. Bar, 20 pixels. cSLO: confocal scanning laser ophthalmoscope; ONC: optic nerve crush. (**c**) Plotting of the Iδ index against compartment width. All the graphs indicate that the mitochondrial distribution was uniform. Two different mice (#1 and #2) were used and the number of axons used for quantification is indicated in the graph. Note that the axons clearly visualized at longer than 120 pixels in length could be analyzed. (**d**) The average density of mitochondria in the axons. The number of axons used for quantification is indicated in the graph. Data are presented as the mean ± SEM. **** *p* < 0.0001, ns = not significant.

**Figure 6 ijms-24-04356-f006:**
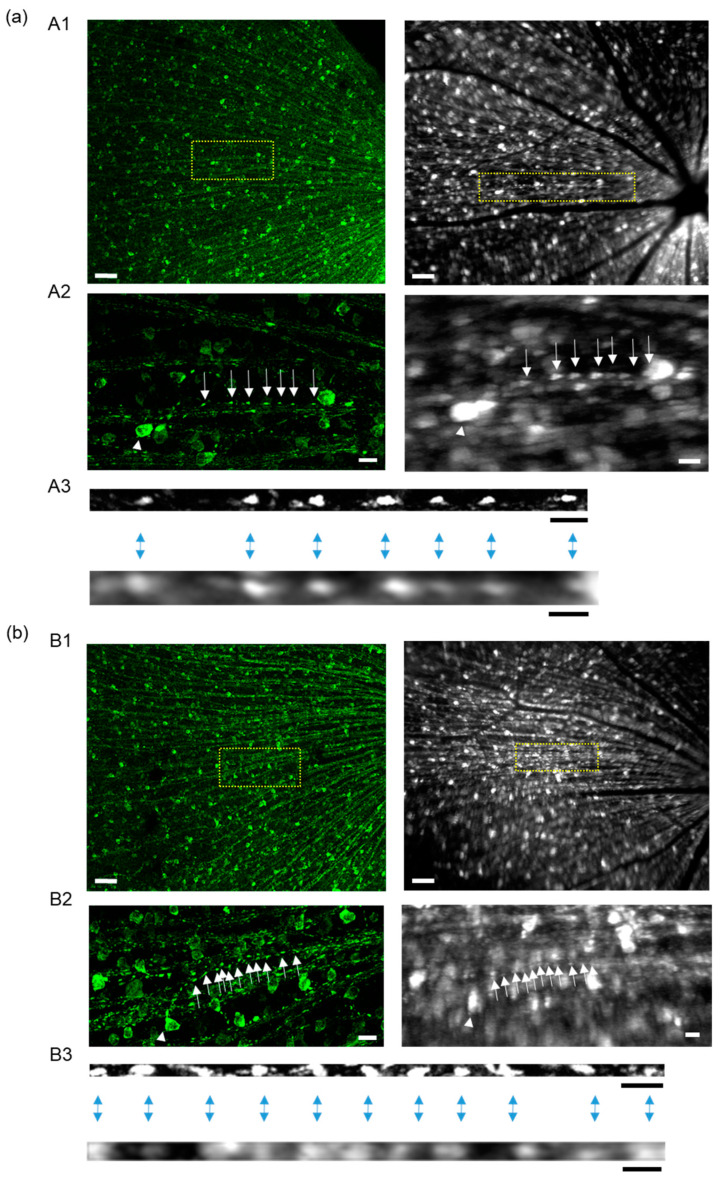
Comparison of mitochondrial distribution within a single axon by in vivo cSLO fundus imaging and by the in vitro imaging of flat-mounted retinal sections with confocal microscopy. (**a**) An example of the comparison between these two images. (**b**) Another example of a comparison. Mitochondria (YFP) shown in green in fundus images taken with cSLO and in white in images of –flat-mounted retinal sections taken with a confocal microscope. Bars, 100 μm in the left panels and 100 pixels in the right panels in A1 and B1, 20 μm in the left panels and 20 pixels in the right panels in A2 and B2, and 10 μm in the left panels and 10 pixels in the right panels in A3 and B3. Images of flat-mounted retinal sections taken with a confocal microscope are exhibited in the left panels in A1 and B1. Fundus images of mitochondria taken with cSLO are exhibited in the right panels in A1 and B1. The yellow squares in A1 and A2 are enlarged in B1 and B2. White arrows and arrowheads indicate a single mitochondrion and cell body, respectively. Mitochondrial distribution in linearized axons in flat-mounted retinal sections captured with confocal microscopy are exhibited in the upper panels of A3 and B3, and those in fundus images captured with cSLO are exhibited in the lower panels in A3 and B3. Mitochondria (YFP) shown in white in A3 and B3. Blue two-headed arrows indicate the corresponding mitochondria in the axon.

## Data Availability

Data sharing is not applicable.

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
