# Peer review of "Optic Nerve Injury Enhanced Mitochondrial Fission and Increased Mitochondrial Density without Altering the Uniform Mitochondrial Distribution in the Unmyelinated Axons of Retinal Ganglion Cells in a Mouse Model"

_ijms, 2023, doi:10.3390/ijms24054356_

Round 1

Reviewer 1 Report

The authors evaluated the alteration of mitochondrial distributions induced by ONC in vivo (by confocal scanning ophthalmoscope) and in vitro (by flat-mount retinal sections) using transgenic mice, and found ONC did not alter the distribution of mitochondrial in axons of RGC, while the density was increased and the size was attenuated. The manuscript is interesting and well-written, I only have some minor concerns:

1.     Line 116: the left panel should be control.

2.     Line 186: figure 4c, the mean of the left and right panel should be clarified.

3.     Line 300: RCG should be RGC.

Author Response

Dear Reviewer 1,

Thank you for reviewing and evaluating our manuscripts. We respond to your comments as follows one by one. We hope that this revision will make the manuscripts better and meet your requirements.

1. Line 116: the left panel should be control.

I corrected as follows,

Left panel: control, middle panel: 1 day after ONC; right panel: 7 days after ONC.

2.     Line 186: figure 4c, the mean of the left and right panel should be clarified.

I added in the manuscript,

Left panel: ONC–operated group, right panel: sham-operated (control) group.

3.     Line 300: RCG should be RGC.

I corrected 'RCG' to 'RGC'.

I also found 'RCG' in line 45, and corrected it.

Sincerely yours,

Takahiro Tsuji

University of Fukui

Department of Ophthalmology, Faculty of Medical Sciences,

University of Fukui

23-3 Shimoaizuki, Matsuoka, Eiheiji, Yoshida, Fukui, 910-1193, Japan

TEL +81-776-61-8403

FAX +81-776-61-8131

Reviewer 2 Report

In this manuscript, the authors reported their investigation about how the mitochondrial changes after ONC surgery. They found that the distribution doesn't changes, however, the density increases while the size decreases, which means ONC induces mitochondrial fission other than fusion. Both confocal microscopy and confocal SLO were used, and it is very impressive that they could correlate the same mitochondrial in these two imaging modalities (Fig. 6). Overall, it is a thorough study. I only have a few minor concerns as follows:

0. Abstract: suggest adding texts in [ ] for clarification.

Line 14: Mitochondria are deeply involved in maintaining [the health of] RGCs and their axons.

Line 22: axons of RGCs that survived [after] the ONC

Line 24: These results [(density increased, size decreased)] suggest that ONC

Line 25: [, which] may help to prevent axonal degeneration 25 and apoptosis.

Line 26: The in vivo visualization system of [ fluorescently labeled ]axonal mitochondria in RGCs may be applicable 26 in the [detection] of [the] progression of optic neuropathy [in animal studies].

1. Suggest including some background knowledge of the percentage of stationary mitochondria and motile mitochondria.

2. Line 91, please give the mathematical definition of Iδ index and the compartment area ratio.

Could you please add a dot or dot range of the experimental results on the left panel of Fig. 1 ©? Or other words, what is the value of the measured compartment area ratio and Iδ index? Please mark it/them on Fig. 1 (c) left panel.

I saw the unit of the compartment area in Fig. 3 (d) is um, I thought the area should correspond to um^2, while um should correspond to e.g. 'length'.

However, the unit becomes pixels in Fig. 5 (c), please keep it consistent.

3. Fig. 2, is it in purpose to use n=6 for '1-day' while n=5 for '7-day' experiments?

The same question for Fig. 3 (e), the n differs quite a lot in these two experiments, why?

4. Line 176, what is 'halation'?

5. Line 178: As time progresses after mice are anesthetized, their lenses become cloudy.

I saw a contact lens was used, so, the cloud was probably due to the low body temperature. With a heating pad + a contact lens, the mouse’s pupil could maintain for up to a few hours[1,2].

[1] In vivo wide-field multispectral scanning laser ophthalmoscopy–optical coherence tomography mouse retinal imager: longitudinal imaging of ganglion cells, microglia, and Müller glia, and mapping of the mouse retinal and choroidal vasculature, Journal of Biomedical Optics, 2015.

[2] Cataract-preventing contact lens for in vivo imaging of mouse retina, BioTechniques, 2018.

6. Fig. 4, please mark 'Operated Eyes' for (a) and (c) left panel, please mark 'Control Eyes' for (b) and (c) right panel.

7. Line 376, please add more information (e.g. diameter, thickness, material) about the contact lens.

Author Response

Dear Reviewer 2

Thank you very much for reviewing and evaluating our manuscripts, entitled ‘Optic Nerve Injury Enhanced Mitochondrial Fission and Increased Mitochondrial Density Without Altering the Uniform Mitochondrial Distribution in the Unmyelinated Axons of Retinal Ganglion Cells in a Mouse Model’ (manuscript ID ijms-2212712). We respond to your comments one by one as attached the file.

We hope that this revision will make the manuscripts better and meet your requirements to be suitable for publication in the Journal.

Sincerely yours,

Takahiro Tsuji

University of Fukui

Department of Ophthalmology, Faculty of Medical Sciences,

University of Fukui

23-3 Shimoaizuki, Matsuoka, Eiheiji, Yoshida, Fukui, 910-1193, Japan

TEL +81-776-61-8403

FAX +81-776-61-8131
